# High Folate, Perturbed One-Carbon Metabolism and Gestational Diabetes Mellitus

**DOI:** 10.3390/nu14193930

**Published:** 2022-09-22

**Authors:** Jessica M. Williamson, Anya L. Arthurs, Melanie D. Smith, Claire T. Roberts, Tanja Jankovic-Karasoulos

**Affiliations:** Pregnancy Health and Beyond Laboratory, Flinders Health and Medical Research Institute, Flinders University, Adelaide, SA 5042, Australia

**Keywords:** folate, vitamin B12, homocysteine, choline, betaine, gestational diabetes mellitus, one-carbon metabolism

## Abstract

Folate is a dietary micronutrient essential to one-carbon metabolism. The World Health Organisation recommends folic acid (FA) supplementation pre-conception and in early pregnancy to reduce the risk of fetal neural tube defects (NTDs). Subsequently, many countries (~92) have mandatory FA fortification policies, as well as recommendations for periconceptional FA supplementation. Mandatory fortification initiatives have been largely successful in reducing the incidence of NTDs. However, humans have limited capacity to incorporate FA into the one-carbon metabolic pathway, resulting in the increasingly ubiquitous presence of circulating unmetabolised folic acid (uFA). Excess FA intake has emerged as a risk factor in gestational diabetes mellitus (GDM). Several other one-carbon metabolism components (vitamin B12, homocysteine and choline-derived betaine) are also closely entwined with GDM risk, suggesting a role for one-carbon metabolism in GDM pathogenesis. There is growing evidence from in vitro and animal studies suggesting a role for excess FA in dysregulation of one-carbon metabolism. Specifically, high levels of FA reduce methylenetetrahydrofolate reductase (MTHFR) activity, dysregulate the balance of thymidylate synthase (TS) and methionine synthase (MTR) activity, and elevate homocysteine. High homocysteine is associated with increased oxidative stress and trophoblast apoptosis and reduced human chorionic gonadotrophin (hCG) secretion and pancreatic β-cell function. While the relationship between high FA, perturbed one-carbon metabolism and GDM pathogenesis is not yet fully understood, here we summarise the current state of knowledge. Given rising rates of GDM, now estimated to be 14% globally, and widespread FA food fortification, further research is urgently needed to elucidate the mechanisms which underpin GDM pathogenesis.

## 1. Introduction

Gestational diabetes mellitus (GDM) is defined as glucose intolerance and subsequent hyperglycaemia diagnosed during pregnancy that is typically resolved post-delivery of the neonate [1]. Over the past two decades, the incidence of GDM has increased globally. In Australia, the incidence of GDM has risen dramatically, tripling from 5% in 2008–2009 to 16% in 2017–2018 [2]. Across the same time frame, the incidence of type 2 diabetes mellitus (T2DM) has remained constant (4.1% in 2008 compared to 4.4% in 2018 [3]), suggesting specific pregnancy adaptations contribute to development of glucose intolerance and subsequent hyperglycaemia. Considering multiple immediate and long-term health consequences to the mother and the baby, there is urgent need to identify factors that are driving the GDM rise, not just in Australia but globally. Over the past decade evidence is mounting on the association between GDM with high intake of folic acid (FA) and high circulating folate during pregnancy. Despite an increasing trend towards higher serum folate levels [4,5,6] and increasing evidence to suggest an association with GDM [7,8,9,10,11,12,13,14,15,16], the mechanistic link between excess folic acid intake and GDM development is unknown. This review will summarise the current knowledge on high FA intake and high circulating folate, the consequences for function of the one-carbon metabolic pathway and the implications for GDM pathogenesis.

## 2. Gestational Diabetes Mellitus (GDM): Health Significance

The risk of adverse perinatal and long-term events associated with GDM complicated pregnancies are well documented for both the mother and her baby. Perinatally this includes increased risk of maternal hypertension, preeclampsia, birth injury and caesarean section [17,18]. Long-term maternal health implications associated with having a GDM-complicated pregnancy include increased risk of developing metabolic syndrome [19,20,21], T2DM [22], and cardiovascular disease [23,24], specifically elevated blood pressure, serum triglycerides, and blood glucose, detected as early as 1 year post-partum [25].

In utero exposure to hyperglycaemia and GDM is associated with increased health risks for the baby, including preterm birth, macrosomia, large for gestational age, shoulder dystocia, neonatal respiratory complications and hypoglycaemia [17,18]. There is also increasing concern in the role of GDM exposure in utero on metabolic programming of the fetus, increasing risk of long-term metabolic dysfunction. A meta-analysis (3 studies, *n* = 4421) demonstrated that in utero exposure to GDM increased risk of metabolic syndrome in offspring (RR 2.07, 95% CI: 1.26–3.42) [21]. Another meta-analysis of 24 studies found offspring of GDM-complicated pregnancies had increased risk of markers of cardiovascular disease in childhood, including elevated systolic blood pressure, BMI and elevated fasting blood glucose [26]. In utero exposure to GDM-complicated pregnancy is also a risk factor for development of childhood diabetes (*p* < 0.001) [27]. Further, maternal glucose concentration during pregnancy had a significant inverse relationship with insulin sensitivity, and a positive relationship with static β-cell response in children aged 5–10 years [28]. A study of adults who were exposed to in utero hyperglycaemia found reduced insulin sensitivity and secretion (*p* < 0.005) [29]. There is increasing evidence that in utero exposure to GDM has long-term metabolic [21,29] and cardiovascular [26] effects on offspring health in later life.

Thus, the adverse maternal and neonatal outcomes associated with GDM are well known but the underlying mechanisms contributing to GDM, are less well understood.

## 3. Gestational Diabetes Mellitus: Risk Factors and the Role of Folic Acid in GDM Aetiology

In Australia, the incidence of GDM has tripled since 2008. Largely the dramatic rise in GDM has been attributed to changes in known maternal risk factors and diagnostic criteria. However, known GDM risk factors, such as maternal age [30], body mass index (BMI) [31] and ethnicity [32], fail to completely explain the rise in GDM incidence in Australia. Advanced maternal age (≥35 years) is a well-known risk factor for GDM [30]. However, the proportion of women giving birth aged ≥35 remained stable from 2008 (22.9% of women giving birth) until 2016 (22.8%), [33] by which time the incidence of GDM had increased to 13.8% [2]. BMI is also a well-established risk factor for GDM, but BMI alone also fails to explain this dramatic rise [31]. While overall BMI is increasing, modest trends are observed amongst women who gave birth. In 2012, 26.5% and 20.7% of women giving birth were overweight or obese, respectively, rising to 26.8% and 21.8% in 2018 [34]. Unfortunately, collection of uniform maternal BMI national data began in 2012, and therefore no pre-FA fortification data exist. It is possible that the modest rise in maternal BMI may contribute to GDM incidence, but it is not likely to explain the three-fold increase observed over the past 12 years. Similarly, the change in diagnostic criteria with the adoption of the International Association of Diabetes in Pregnancy Study Groups guidelines from 2014, also fails to account for the increase, given GDM cases had already more than doubled by 2014 (12%) and have continued to rise since [2,35]. Population ethnicity, maternal age, BMI and changing diagnostic criteria likely contribute to rising GDM, but even combined are unlikely to explain a three-fold increase in GDM across the past decade. Interestingly, the rise in GDM closely follows the introduction of mandatory FA food fortification in Australia, implemented in 2009, suggesting a potential mechanistic role of FA in GDM pathogenesis. 

### 3.1. High FA and GDM Risk

Emerging evidence increasingly links high FA intake with GDM (Table 1). A 2016 prospective cohort study (*n* = 1938) found an increased risk of GDM associated with daily FA supplementation in the first trimester (aOR: 2.25 95% CI: 1.35–3.76) noting an additive effect of pre-pregnancy BMI ≥ 25 kg/m^2^ on further increasing GDM risk (OR: 5.63 95% CI: 2.77–11.46) [7]. More recently, the Shanghai Preconception Cohort Study found FA supplementation increased GDM risk (aOR: 1.73 95% CI: 1.19–2.53, *p* = 0.004) [8]. The authors further showed that RBC folate exceeding 600 ng/mL (1360 nmol/L) is associated with an increased GDM risk (aOR: 1.58 95% CI: 1.03–2.41, *p* = 0.033). Duration of FA supplementation has also been implicated with GDM. FA supplementation for ≥3 months prior to pregnancy is associated with an increased risk of GDM, compared to supplementation < 3 months (aRR: 1.72 95% CI: 1.17–2.53, *p* < 0.01) [9] and <2 months (aOR: 3.45 95% CI: 1.01–11.8, *p* < 0.05) [10]. 

Higher red blood cell (RBC) folate was detected in GDM pregnancies (*n* = 392) compared to controls (*n* = 1890) in the second trimester across each quintile (*p* trend = 0.012) [11]. This is consistent with a 2021 study of pregnant women that found RBC folate prior to 12 week’s gestation was indicative of increased GDM risk (aOR: 2.473 95% CI: 1.013–6.037, *p* = 0.047) [12]. Similarly, a study of women between 24–28 weeks’ (*n* = 406) found increased serum folate was positively associated with increased GDM risk (OR 1.98 95% CI: 1.00–3.90, *p* = 0.049) [14]. Concordantly, a prospective study (*n* = 4746) found increased serum folate prior to 16 weeks’ gestation was associated with an 11% higher relative risk of GDM after adjusting for confounding variables (aRR: 1.11 95% CI: 1.036–1.182, *p* = 0.002) [15]. A nested case–control cohort similarly found that maternal serum folate levels at 15 weeks’ gestation were higher in women who developed GDM (*n* = 33, mean ± SD: 37.6 ± 8 nmol/L) compared to uncomplicated pregnancies (*n* = 111, 31.9 ± 11.2, *p* = 0.007) but statistical significance was lost after adjusting for maternal age, BMI and smoking status [16]. 

Collectively, there is increasing evidence for a role for FA in GDM epidemiology. To understand how changes to maternal FA intake and circulating folate can contribute to GDM it is imperative to understand the role of folate and its synthetic derivative FA in pregnancy health.

### 3.2. High FA and Metabolic Dysfunction

The mechanistic relationship between high FA and GDM risk is poorly elucidated. A growing body of research suggests high FA exposure in utero may program metabolic function in offspring, though the maternal effects are less well studied. In human studies, higher maternal folate has also been associated with insulin resistance [36,37], and adiposity [37] in offspring. This is consistent with findings from murine models, which suggest maternal FA intake is associated with adverse metabolic outcomes in the offspring, specifically adipocyte morphology [38], glucose intolerance and insulin resistance [39], and impaired insulin synthesis and fat metabolism [40]. The underlying molecular pathways for altered adiposity and increased insulin resistance due to high FA are currently unclear, although altered DNA methylation, upregulation of lipogenesis pathway genes and downregulation of glucose transporter 4 (GLUT4) in adipose and muscle tissues have been implicated by these studies. Other studies have also found maternal high fat and high folate diets interact to exacerbate disturbed lipid metabolism [41,42] and insulin signalling [42] in offspring. There is significant evidence for a role of high FA in metabolic programming in the offspring but a role in maternal metabolism is yet to be demonstrated. One study assessed metabolic function in a study of male Sprague Dawley rats fed either high fat diet or low-fat diet with either excess FA (7.5 mg FA/kg) or control FA (0.75 mg/kg) for 12 weeks [43]. In high fat-treated rats, excess FA resulted in increased adipocyte size and induction of several lipogenic genes in adipose tissue and impaired glucose tolerance [43]. Interestingly, in low fat diet-treated mice, excess FA did not alter body weight or composition, nor glucose tolerance [43]. Theoretically, high FA may have adverse maternal metabolic effects, either in isolation or in combination with a high fat diet. A potential relationship between concomitant high fat and high FA diets has significant implications in pregnancy health, particularly in explaining a causal relationship with GDM. However, further research is needed. 

### 3.3. High FA and β-Cell Dysfunction

High FA has been shown to alter β-cell function via the folate receptor α (FOLRα) signalling [44,45]. One study reports that an optimal dose of FA (0.1 μM) promotes differentiation of porcine pancreatic stem cells into insulin-secreting cells, as well as increasing cell viability and proliferative capacity. However, the highest FA dose (1 μM) reduced cell viability and proliferation compared to the optimal dose [44]. The authors propose the mechanism of FA action in porcine pancreatic stem cells occurs via FA-FOLRα binding. Interestingly, FOLRα expression was also upregulated in response to 0.1 μM FA and downregulated in response to 1 μM treatment. This research suggests a mechanistic role for high dose FA in β-cell dysfunction in a mammalian model. However, further research is needed to confirm conservation to human models, specifically with reference to the adaptive β-cell response that is necessary for pregnancy [46] and implicated in GDM pathogenesis [47,48].

## 4. Folate and One-Carbon Metabolism in Pregnancy

Folate is a dietary micronutrient essential to one-carbon metabolism, a universal process comprised of multiple interconnected pathways; specifically, the folate cycle, the methionine cycle, and the trans-sulphuration pathway (Figure 1) [49,50]. One-carbon metabolism is critical for the biosynthesis of purines, thymidylate and re-methylation homocysteine to methionine. Homocysteine is a proinflammatory amino acid and high levels can be detrimental to health, including pregnancy [51]. Thus, perturbations to one-carbon metabolism can impact maternal and neonatal health [52,53,54,55]. Humans require dietary folate to maintain different functional aspects of the one-carbon metabolic pathway, whether in the form of non-synthetic folates, obtained from food such as dark leafy greens, eggs, and liver or acquired from FA, a synthetic form, obtained from fortified foods or supplements [56,57]. Unlike non-synthetic folate, FA requires reduction via dihydrofolate reductase (DHFR) to dihydrofolate (DHF) and sequentially tetrahydrofolate (THF) [58]. THF is then interconverted to intermediate metabolites 10-formylTHF, 5,10-methenylTHF and 5,10-methyleneTHF, before incorporation into the folate cycle as the most reduced folate form 5-methylTHF [59]. Expression of one-carbon metabolic enzymes is shown in all human adult tissues suggesting that all tissues should be able to generate de novo one-carbon units. Folates carrying one-carbon units do not freely transport across intracellular membranes. Thus, a complete set of one-carbon enzymes exist in both the mitochondria and the cytosol to synthesize 5,10-methylene-THF in both compartments. It is believed that the one-carbon metabolic pathway is localized in the mitochondria to uncouple this pathway from glycolysis [60,61].

Physiologically, folate has essential functions in growth, differentiation and repair, and thus adequate supply is highly critical during fetal and placental development [62,63]. The human newborn has been estimated to have grown from one to 1.25 × 10^12^ cells during gestation with an additional 0.25 × 10^12^ cells in the umbilical cord, placenta and fetal membranes [64,65]. Folate plays critical physiological roles in pregnancy to support uterine and placental growth and development, sustained cell division necessary for fetal growth, and specifically in prevention of neural tube defects (NTDs) [66,67,68]. Further, preconception folic acid intake has also been positively associated with fetal growth and reduced risk for small-for-gestational-age (SGA) [69] and reduced risk of spontaneous preterm birth (sPTB) [70]. There is also some evidence for a modest protective effect of folic acid supplementation in preeclampsia risk [71]. In pregnancy, folate requirements increase 5- to 10-fold, to accommodate the needs of the growing feto-placental unit, as well as increasing maternal metabolic demands [66]. Currently, guidelines recommend 400 μg daily FA intake to prevent NTDs [72]. As dietary folate is often inadequate [73], many countries (~92) have mandatory FA food fortification policies, as well as recommendations of periconceptional FA supplementation [74].

Mandatory FA food interventions have significantly reduced folate deficiency and the occurrence of neural tube defects, most recently reviewed by Wilson & O’Connor (2021) [75]. Indeed, in countries with FA food fortification and FA supplementation guidelines for pregnancy, folate excess is more common than folate deficiency. A systematic review confirms that in the context of mandatory FA fortification, most women preconception, and through pregnancy, are exceeding the upper tolerable limit (~1000 μg/d) [76]. While direct consequences of nutrient deficiencies have been well documented [77,78,79], the effects of excess FA intake are less well elucidated and are more likely to have direct effects on one-carbon metabolism and ultimately complicated and indirect effects on maternal and fetal health, and pregnancy outcome.

## 5. High FA Intake and Unmetabolized FA (uFA)

High intake of FA is known to increase circulating unmetabolized FA (uFA), and this is becoming increasingly prevalent, particularly in the context of food fortification. uFA has been detected in the serum in non-pregnant [6] and pregnant [80] populations, umbilical cord blood [81,82,83], neonatal plasma [84], and in breast milk [85,86,87]. uFA has also been detected in maternal and neonatal plasma [88], serum [82] and umbilical cord blood [82] in a voluntary fortified population.

Accumulation of uFA itself may be detrimental, though there is limited research regarding the adverse effects of circulating uFA in pregnancy. Studies of allergic disease [89] and neurodevelopment [90], have found no adverse effects associated with uFA. However, some research indicates that higher maternal folate is associated with insulin resistance [36,37] and adiposity [37] in offspring. A direct role for circulating uFA in impairing maternal metabolism or in GDM pathogenesis remains to be elucidated. While there is growing evidence for the presence of uFA, the extent to which maternal uFA may be harmful is not well established, nor has a relationship between uFA and pregnancy outcome, including GDM, been described.

The presence of circulating uFA is largely attributed to excess intake, limited enzyme metabolic capacity and a low saturation threshold [91,92]. Understanding the relationship of uFA and one-carbon metabolism remains a significant area for future research, particularly considering the increasing prevalence of uFA, and mounting evidence to suggest a relationship between high FA and GDM.

## 6. Effects of Excess FA on the Players in the 1C-Metabolism Pathway

### 6.1. Excess FA Saturates Limited DHFR Capacity

The human gastrointestinal tract can efficiently convert non-synthetic folates to 5-methyltetrahydrofolate (5-methylTHF), the active form for methylation pathways, but has a reduced capacity to convert FA [91,93]. Unlike naturally occurring folate, which is converted to THF without requiring the action of DHFR, FA is inactive until DHFR action reduces it sequentially to DHF and then activated THF in a two-step enzymatic process (Figure 2) [94]. DHFR activity in humans is limited and variable [5]. Oral doses of 260–280 μg FA are sufficient to result in unmetabolized circulating FA (uFA) due to saturated DHFR capacity [95]. Mandatory fortification initiatives and periconceptional FA supplementation have resulted in high FA intake exceeding the upper tolerable limit (~1000 μg/d), and thus, high circulating folate levels in pregnancy [96].

### 6.2. Excess FA Reduces MTHFR Protein, Causing a Pseudo-MTHFR Deficiency

After conversion of FA to THF, THF is converted by the methylenetetrahydrofolate reductase (MTHFR) enzyme to 5-methylTHF (Figure 1 and Figure 2), the bioavailable form that acts as an essential co-substrate in homocysteine re-methylation. MTHFR is therefore essential in regulating folate bioavailability.

In murine studies, high dietary FA has been shown to induce MTHFR dysfunction. Dietary FA exceeding recommendations (2 mg/kg mouse chow [97]) by 10-fold (10x-FA, 20 mg FA/kg mouse chow), reduced MTHFR protein in the maternal liver (*p* < 0.05) [98]. Furthermore, dietary FA exceeding recommendation five-fold (5x-FA, 10 mg FA/kg) reduced MTHFR protein [99,100] and activity [100] in the maternal liver, effectively inducing a pseudo-MTHFR deficiency. In addition to FA, DHF has also been characterised as an inhibitor of MTHFR activity in vitro studies [101]. An important caveat is species-specific differences in folate metabolism. Specifically, hepatic DHFR is significantly lower in humans compared to other animals [93,102]. The metabolic rate of DHFR activity in human liver is proposed to be 56 times slower than rat liver [93]. Thus, limited conclusions can be drawn from animal studies. Evidence for MTHFR-deficiency in response to excess FA is well established in mouse models. A pseudo-MTHFR syndrome stemming from excess intake, in the absence of MTHFR polymorphisms, has been documented clinically [103]. Whether this is widely applicable to pregnant women, and its biological significance, is not yet established.

### 6.3. FA Reduces Methionine Synthase (MTR) Activity and Favours the Thymidylate Synthase (TS) Cycle

Paradoxically, FA excess has been implicated in inhibition of folate-dependent pathways, particularly the methionine synthase (MTR) cycle [104,105], instead promoting thymidylate synthase (TS) cycle activity [104]. MTR activity, required for homocysteine re-methylation, is dependent on adequate vitamin B12 availability. The active form of folate, 5-methylTHF, is converted from 5,10-methyleneTHF, however, 5,10-methyleneTHF can alternatively be used as a one-carbon donor in the conversion of deoxyuridine monophosphate (dUMP) to deoxythymidine monophosphate (dTMP) via thymidylate synthase (Figure 1) with key functions in pyrimidine biosynthesis. In a study of *Caenorhabditis elegans (C. elegans)*, 100 mM synthetic FA decreased methylenetetrahydrofolate reductase *(mthf-1)* and methionine synthase *(metr-1)* mRNA but increased thymidylate synthase mRNA [104]. An earlier study of *C. elegans* also demonstrated reduced enzymatic activity of methionine synthase by ~40% and reduced expression of methionine synthase reductase (*mtrr-1*) and *mthf-1* mRNA in high-FA conditions (8.8 μmol/plate) [105]. Taken together, there is emerging evidence that conditions of excess FA favour TS metabolism, facilitating DNA synthesis and cellular growth, at the expense of MTR, effectively reducing MTR action required to support methylation processes [104,105]. Given a secondary action of TS activity is DHF generation, this may further potentiate DHFR saturation induced by increased FA intake

The implications of these perturbations to the one-carbon metabolic pathway are not yet known. An important caveat is that this research is limited to *C. elegans* and has not been assessed in in vivo in humans, ex vivo in tissue, nor in placental cell lines. Favouring TS activity at the expense of cobalamin-dependent MTR activity due to excess FA, becomes increasingly of interest considering a proposed high folate/low B12 interaction in GDM pathogenesis. Whether this phenomenon occurs in the human placenta and whether this may contribute to pregnancy complications, is an interesting area for future research.

### 6.4. Dietary FA Alters Choline and Betaine Metabolism

Disruption of folate metabolism results in use of choline-derived betaine, an alternative methyl donor in methionine synthesis as described above (Figure 1) [106]. This process works in a folate-independent manner to facilitate generation of S-adenosylmethionine (SAM), the universal methyl donor [107]. Following methylation of DNA and other molecules, SAM is converted to S-adenosylhomocysteine (SAH). Therefore, decreases in SAM:SAH ratio indicate reduced methylation potential, whereas reductions in betaine concentrations suggest alternate utilisation as a methionine cycle methyl donor. In a murine study dietary FA exceeding the recommended dietary FA by five-fold, decreased maternal plasma betaine (*p* = 0.043) and increased SAM (*p* = 0.009) but reduced placental SAM:SAH (*p* = 0.039) and SAM (*p* = 0.021) with the latter further exacerbated in male-bearing pregnancies (*p* = 0.006) [100]. This study suggests that high FA may disrupt the maternal methionine pathway, but further research is needed to validate this finding. The consequences of disrupting this pathway, particularly in early gestation hypomethylated placenta, for pregnancy health and outcome, including GDM pathogenesis, need to be established. [108]. In the context of human pregnancy this may promote placental and fetal growth and perturb spatio-temporal methylation of DNA and other molecules important in key developmental steps.

## 7. Association between One-Carbon Metabolism Players and GDM

### 7.1. High FA and Low Vitamin B12 Are Associated with Increased GDM Risk

Beyond adequate folate intake, one-carbon metabolism is also dependent on adequate supply of vitamin B12. Vitamin B12, or cobalamin, is a water-soluble B-vitamin that is an essential cofactor in one-carbon metabolism, and is largely obtained from animal products, making vegetarianism a risk factor for its deficiency [109,110]. There is growing evidence suggesting a role for vitamin B12 insufficiency in GDM development, either in isolation or in conjunction with excess FA intake, summarised in Table 2 [14,15,111,112,113,114]. In a UK-based retrospective case–control study (*n* = 344), women with B12 deficiency (<150 pmol/L) in 2nd and 3rd trimesters had an increased likelihood of GDM diagnosis compared to B12-replete women (aOR: 2.59 95% CI: 1.35–4.98, *p* = 0.004), after adjustment for serum folate, age, parity, smoking status, and ethnicity. Interestingly, folate level did not differ between GDM and non-GDM pregnancies but only folate deficiency, not excess, was reported [115]. The relationship between vitamin B12 deficiency and GDM diagnosis was still present, but attenuated, after adjustment for maternal BMI (aOR: 2.05, 95% CI: 1.03–4.10, *p* = 0.04).

Similarly, a prospective cohort study, found an inverse relationship between vitamin B12 and GDM risk (aOR: 0.856; 95% CI 0.786–0.933; *p* = 0.0004) although this was mediated in part by BMI [15]. Vitamin B12 deficiency (<220 pmol/L in serum) also increased GDM risk (aRR: 1.383, 95% CI: 1.157–1.652, *p* = 0.0004). However, low vitamin B12/high folate was associated with the greatest increase in GDM risk (aRR: 1.742 95% CI: 1.226–2.437, *p* = 0.003) [15]. Interestingly, high folate remained associated with increased risk of GDM after adjustment for vitamin B12 status (aRR 1.11 95% CI: 1.036–1.182, *p* = 0.002) [15]. Similarly, a prospective study at 26 weeks’ gestation (*n* = 913) found the greatest risk of GDM was associated with B12 insufficiency (<150 pmol/L) paired with high folate levels (OR: 1.97 95.3831% CI: 1.05–3.68) [112]. A cross-sectional study of women (*n* = 406, 24–28 weeks’ gestation) found higher folate:B12 ratio was associated with increased GDM risk (OR: 3.08 95% CI: 1.63–5.83) [14]. A small study of vitamin B12-deficient women at 30 weeks’ gestation (*n* = 29) found GDM incidence increased across folate tertiles after adjustment (5.4% (*n* =  7), 10.5% (*n* = 12), 10.9% (*n* = 10), from lowest to highest tertile, *p* =  0.04) [111]. Given vitamin B12 is an essential cofactor in one-carbon metabolism, and the increasing evidence for a role of folate in GDM pathogenesis, it is unsurprising there is also a role for vitamin B12 deficiency in GDM. This is potentially concerning given only food fortification with FA, but not vitamin B12, is mandatory in many countries, especially in the context of widespread vegetarianism, for example in India. The mechanism by which high folate and low vitamin B12 interact to confer GDM risk requires further research. One possible explanation could be the shift from methionine to folate pathways which favour DNA synthesis over methylation and epigenetic programming, leading to not only altered gene expression but increased homocysteine and associated systemic inflammation and insulin resistance.

### 7.2. Circulating Homocysteine Is Elevated in GDM-Complicated Pregnancies

Homocysteine is a proinflammatory, methionine-derived amino acid, essential in one-carbon metabolism [117]. Homocysteine can undergo transulphuration or can be re-methylated to methionine for cellular methylation assuming sufficient folate is available [117]. The state of increased inflammation, such as that associated with hyperhomocysteinemia, is associated with increased insulin resistance [118]. Homocysteine can impair the translocation of certain glucose transporters to the plasma membrane reducing systemic glucose uptake. Thus, high FA intake which can result in impaired methionine cycle activity mimicking folate deficiency, as mentioned above, can lead to hyperhomocysteinemia, increased insulin resistance and reduced glucose uptake.

A systematic review and meta-analysis (12 studies, GDM *n* = 712, control *n* = 1277) indicates homocysteine is significantly elevated in women with GDM-complicated pregnancies, compared to healthy controls (Standard mean difference (SMD) = 0.55; 95% CI: 0.25–0.85, *p* = 0.0003) [119]. Interestingly, two studies stratified participants into three groups: normoglycemic (normal response to 1 h glucose challenge, no oral glucose tolerance test (oGTT) follow-up), glucose intolerant (a positive response to 1 h glucose challenge but normal oGTT results) and GDM [120,121]. Of note, each study had slightly different thresholds for oGTT follow-up, Tarim et al. required 1 h-post challenge blood glucose exceeding >7.5 nmol/L, and Guven et al. required blood glucose exceeding 7.8 nmol/L. Tarim et al. found homocysteine was elevated in women with glucose intolerance and with diagnosed GDM (*p* < 0.001) compared to women with a normoglycemic pregnancy [120]. Guven et al. found a significant (*p* < 0.01) increase in serum homocysteine between women with GDM-complicated pregnancies and normoglycemic pregnancy, but no statistically significant differences between any other group [121]. Interestingly, neither study observed differences in serum vitamin B12 nor folate. Several studies which did not stratify for glucose intolerance have also observed increased serum homocysteine in women with GDM-complicated pregnancies compared to non-GDM pregnancies [122,123,124,125,126].

Conversely, several studies have found no difference in homocysteine concentrations in women with GDM-complicated pregnancies compared to controls. Idzior-Waluś et al. noted no difference in serum homocysteine between women with GDM-complicated pregnancies (*n* = 44, 8 ± 2.0 μmol/L) and non-GDM pregnancies (*n* = 17, 7.4 ± 1.1 μmol/L) between 26 and 32 weeks’ gestation [127]. However, all patients were recruited from an outpatient diabetic clinic, and had all been referred based on positive response to 50 g glucose load; thus, there is likely to be some degree of glucose intolerance within the non-GDM group. Similarly, in a prospective study of pregnant women between 24–28 weeks’ gestation, serum homocysteine did not differ between GDM-complicated pregnancies (*n* = 60, 7.41 μmol/L ± 2.61) and uncomplicated controls (*n* = 19, 8.02 μmol/L ± 2.27) [113]. It is important to note that neither of these studies stratified for glucose intolerance below the threshold for GDM diagnosis, both have very few non-GDM controls, and show homocysteine levels that are higher than is characteristic of non-diabetic pregnancy at a similar gestation [128]. Interestingly, one study at ~34 weeks’ gestation, found reduced total plasma homocysteine in women with glucose intolerance (*n* = 18, 5.0 μmol/L ± 1.7) compared to normoglycemic women (*n* = 190, 6.6 ± 2.0, *p* = 0.024), but no difference between GDM-complicated (*n* = 17, 6.8 ± 2.7) and normoglycemic pregnancies. The authors note that all women with GDM were treated with a low glycaemic diet, which is likely to explain this observation [129]. Similarly, Akturk et al. found no differences in homocysteine levels in late pregnancy (32–39 weeks’), though all women diagnosed with GDM (*n* = 54, according to American Diabetes Association criteria, at 24–28 weeks’ gestation [130]) were treated following GDM diagnosis (*n* = 48 diet only, *n* = 6 diet and insulin) [131].

Together, these studies (Summarised in Table 3) suggest that homocysteine levels are elevated in GDM-complicated pregnancies in mid-late gestation. Furthermore, two studies which found no differences in homocysteine in late pregnancy also involve treatment, suggesting elevated circulating homocysteine in GDM are modifiable [129,131]. Predominantly, these studies assess homocysteine in the second and third trimester of pregnancy, after GDM onset, making it difficult to assess causation between homocysteine and GDM pathogenesis. Only one study measured fasting serum homocysteine in early gestation (8–12 weeks’ gestation), finding no difference in homocysteine levels between normoglycemic (*n* = 83, 14.41 ± 7.98 μmol/L) and GDM-complicated pregnancies (*n* = 7, 15.66 ± 7.61, *p* = 0.6312) [132]. However, the low statistical power of this study and very high homocysteine levels of this cohort, mean that limited conclusions can be drawn. The high homocysteine levels in this cohort may be somewhat explained by widespread vegetarianism and associated vitamin B12 deficiency, which is common in the study country, India [133]. However, the authors do not provide patient data to confirm this speculation. Furthermore, vitamin B12 and folate status that are known to influence homocysteine were not assessed. It remains unclear whether homocysteine contributes to GDM aetiology or is a consequence of hyperglycaemia. In addition, whether elevated homocysteine reflects perturbation of other one-carbon metabolites or whether high folate and high homocysteine act independently in GDM pathophysiology remain unknown. Further research in early pregnancy is needed to elucidate the nature of the homocysteine-GDM relationship.

There are several mechanisms by which high homocysteine may contribute to GDM:

Via pancreatic β-cells: Pancreatic β-cells secrete insulin to maintain glucose homeostasis [134]. During pregnancy, there is an adaptive increase in β-cell mass to increase insulin release, accommodating the increased insulin resistance of pregnancy [46,48,135]. Insufficient β-cell adaptation is associated with hyperglycaemia and GDM diagnosis [136,137]. In vitro, homocysteine has been shown to inhibit glucose-induced insulin secretion by β-cells in a dose-dependent manner at moderate (5.6 mM, *p* < 0.001) and stimulatory (16.7 mM, *p* < 0.001) glucose concentrations [138,139]. Serum homocysteine was inversely associated with several parameters of pancreatic islet β-cell function, in T2DM patients but a similar effect in GDM has not yet been characterised [140]. Further research is needed to clarify the relationship between elevated homocysteine and β-cell function in GDM pathogenesis.

Via the placenta to induce oxidative stress: Within the placenta, oxidative stress is necessary for angiogenesis, immunoregulation and vasoactivity [141]. However, an imbalance of reactive oxygen species and antioxidant activity can result in elevated oxidative stress and lipid peroxidation, and subsequent cellular damage [142,143]. The role of homocysteine as a pro-oxidant is well-established [144,145,146,147], as is the role of oxidative stress in insulin resistance [118,148] and disruption of insulin signalling [149,150].

In *C. elegans,* malondialdehyde (MDA), a marker of lipid peroxidation, and hydrogen peroxide (H_2_O_2_), an activator of oxidative stress, were increased after high dose-FA induced elevated homocysteine [105]. Other research indicates that supraphysiological FA can exacerbate lipid peroxidation in oxidative stress conditions [151]. In BeWo cells (choriocarcinoma cell line) oxidative stress was induced with tert-butylhydroperoxide (TBH) treatment under deficient (1 nM), physiological (20 nM) and supraphysiological (2.3 μM) FA treatment. The latter increased MDA content compared to physiological FA, both in response to 100 μM and 300 μM TBH treatment [151]. However, the interaction between FA, homocysteine and oxidative stress requires further research. Both oxidative stress [152,153,154,155] and lipid peroxidation [156,157] markers, have been shown to be associated with pregnancies complicated by GDM. In addition, there may also be a role for excess FA in inducing or exacerbating existing oxidative stress, either through elevated homocysteine or an alternate pathway. Further research is warranted to elucidate the mechanisms by which elevated homocysteine, oxidative stress, and potentially uFA contribute to GDM pathogenesis.

Via the placenta to induce apoptosis: Homocysteine can also cause placental dysfunction by promoting cellular apoptosis. In vitro homocysteine treatment can induce trophoblast apoptosis in primary human placental trophoblasts (36 weeks’ gestation) [158]. Interestingly, a follow-up study found that treatment with FA (20 nmol/L, a plasma concentration considered healthy), alleviated homocysteine-induced apoptosis [159]. Similarly, in primary cultured human trophoblasts collected at term, homocysteine-thiolactone, a homocysteine oxidation product, induced apoptosis in a dose-dependent manner [160]. In contrast, treatment with folate (10 μmol/L, representing a supraphysiological dose) did not alleviate homocysteine-induced apoptosis

Despite some evidence that homocysteine induces trophoblast apoptosis, a role for apoptosis in GDM remains controversial. There is both evidence for an increased [161,162] and decreased [163] trophoblast apoptotic index in placentae from GDM-complicated pregnancies, compared to those from uncomplicated pregnancies. Further research to definitively demonstrate the role of trophoblast apoptosis in GDM is needed, with an emphasis on potential effects of homocysteine and FA.

Via dysregulated hCG secretion: A fourth mechanism by which high homocysteine levels may contribute to GDM is through altering hormone profiles, specifically human chorionic gonadotrophin (hCG). hCG is a hormone produced by placental syncytiotrophoblasts and is critical to pregnancy maintenance [164]. A systematic review and meta-analysis concluded that first trimester β-hCG is reduced in women subsequently diagnosed with GDM, compared to women who remain normoglycemic [165]. In placental explants collected from uncomplicated pregnancies, in vitro homocysteine treatment significantly reduced hCG secretion at 20, 40 and 80 μmol/L (50–80% reduction, *p* < 0.004) [158]. Similarly, in placental villous trophoblasts, isolated from uncomplicated term placentae, a 43% reduction of hCG secretion under 20 μmol/L homocysteine treatment (*p* < 0.02) was observed. FA treatment (40 nmol/L) was able to restore hCG secretion (*p* < 0.05) [159], though the effects of excess FA was not evaluated. Ahmed et al., observed no differences in hCG secretion in placental explants nor in BeWo villous cytotrophoblast, under FA deficient (2 ng/mL), physiological (20 ng/mL), elevated (200 ng/mL) and supraphysiological (2000 ng/mL) conditions [166]. This suggests FA may not directly alter trophoblastic hCG secretion, instead its’ action may be mediated through elevating homocysteine. A caveat of the existing work is the use of term rather than early gestation placentae, which makes it difficult to ascertain a causal relationship in GDM pathogenesis. Current knowledge indicates that homocysteine reduces hCG secretion and reduced hCG has been previously implicated in GDM-complicated pregnancies [165]. Further research is needed to clarify whether elevated homocysteine contributes to GDM pathophysiology or is just an indicator of perturbed one-carbon metabolism.

### 7.3. Interactions between Excess Folate, Vitamin B12, Homocysteine and Risk for GDM

Given folate, vitamin B12 and homocysteine have all been previously implicated in GDM risk, and each is necessary for one-carbon metabolic function, the complex interplay of these metabolites becomes increasingly relevant. Remethylation of homocysteine to methionine occurs via cobalamin-dependent enzyme MTR, with 5-methylTHF as a necessary cofactor in MTR activity [53]. Given Vitamin B12 is essential in 5-methylTHF uptake, insufficient B12 can result in folate becoming trapped in this form [56,167]. Thus, the methionine pathway is both highly dependent on adequate folate availability and B12 uptake, as well as MTR action [56,105]. Generally, homocysteine has an inverse relationship with folate, with elevated homocysteine is considered to be a sensitive marker of folate deficiency [168,169]. Given both high homocysteine and high FA are associated with GDM [11,13,14,15,16] it is plausible the homocysteine remethylation pathway is perturbed in both deficient and high FA conditions, but the nature of this relationship requires further investigation. Some evidence suggests excess FA may directly induce elevated homocysteine. In both BeWo and JEG3 human choriocarcinoma cell lines, increased homocysteine was observed under supraphysiological FA (2000 ng/mL) conditions compared to normal physiological (20 ng/mL) treatment [170]. High dose FA supplementation in *C. elegans* has been shown to induce high homocysteine [105]. However, there is limited in vivo human research to verify the effect of supraphysiological FA and further investigation is required.

Evidence suggests the FA-homocysteine relationship may be mediated through vitamin B12 deficiency. In a study of healthy adults, homocysteine concentrations only decreased across increasing serum folate categories when vitamin B12 was >148 pmol/L (*p* < 0.001), and this was not observed when vitamin B12 was <148 pmol/L [171]. Beyond the vitamin B12-folate interaction observed clinically, there is also in vitro evidence that vitamin B12 sufficiency is necessary to reduce elevated homocysteine levels. In BeWo and JEG3 choriocarcinoma cell lines, treatment with various forms of vitamin B12, including cobalamin and combined methylcobalamin (MeCBl) and adenosylcobalamin (AdCbl), lowered homocysteine levels, which were elevated by supraphysiological FA treatment [170]. This suggests that rather than a direct role of high FA intake in perturbing homocysteine remethylation, vitamin B12 instead acts as the limiting factor. However, FA may have a direct, concomitant role in interfering with vitamin B12-dependent metabolism through reduced MTR action [104,105], which is exacerbated in cases where vitamin B12 is already limited.

### 7.4. Choline-Derived Betaine Is Associated with Decreased GDM Risk

Choline is an essential nutrient, that can be synthesized endogenously in the human liver but obtaining additional dietary choline from sources such as red meat, poultry, fish, eggs and soybeans, is necessary to achieve adequate levels in circulation [172]. Dietary choline can be either water-soluble, which is metabolized in the liver, or lipid-soluble and transported through the lymphatic system. While choline acts in multiple pathways, in one-carbon metabolism choline is converted to betaine for DNA methylation, phospholipid synthesis and fetal neurodevelopment [173].

When folate or vitamin B12 supply are limited, choline-derived betaine is used as a methyl donor for homocysteine re-methylation and global methylation processes [174] (Figure 1). The role of choline and betaine in GDM development has been understudied, a summary of existing research is provided in Table 4. Two studies report a protective role of maternal betaine in GDM [175,176]. In a prospective cohort study (*n* = 486), maternal plasma betaine (≤200 nmol/mL) in early pregnancy (median: 10 weeks’ gestation, IQR: 9–11) was associated with increased GDM risk (OR: 5.00 95% CI: 2.76–9.07) [175]. Similarly, in a cohort study of dichorionic twin pregnancies (*n* = 187), plasma betaine had an inverse relationship with GDM risk (*p* trend = 0.015) [176].

Conversely, one study found no association between maternal circulating betaine and GDM risk [177]. However, plasma betaine was decreased in cord blood of neonates of GDM-complicated pregnancies (18.5 ± 3.9 μmol/L) compared to uncomplicated (21.2 ± 4.7 μmol/L, *p* = 0.02). Given there were no differences in maternal betaine observed, the authors proposed that GDM may alter the transfer of betaine to the fetus and/or that women with a GDM-complicated pregnancy utilise betaine for maternal metabolism, resulting in depleted neonatal cord concentrations [177]. Collectively, there is emerging evidence that choline-derived betaine may play a protective role in GDM risk. Further research is needed to determine the relationship of choline and betaine in GDM pathogenesis, particularly with reference to other one-carbon metabolites.

While it has been established that choline can modulate important markers of placental function [178], the mechanism by which insufficient choline can increase GDM risk is not well understood. The oxidation of choline to betaine occurs in the mitochondria, and is catalysed by choline dehydrogenase (CHD) which, based on animal studies, is presumed to be located on the inner mitochondrial membrane [179].The necessity of effective mitochondria in betaine conversion suggests perturbed mitochondrial function may be the mechanistic link. Reduced mitochondria content [180] and alterations to placental mitochondrial dynamics and metabolism, specifically increased mitochondrial fusion [181], have been observed in the placenta of GDM-complicated pregnancies. Further, Abbade et al. propose hyperinsulinemia mediates the observed mitochondrial dysfunction using JEG-3 cells as an in vitro model. The relationship between mitochondrial function and GDM needs further clarification to establish whether mitochondrial modifications underpin GDM pathogenesis, or are a consequence of the GDM state, specifically the accompanying hyperinsulinemia. Further establishing the interplay between choline-betaine conversion in the pathway is also of significant interest, given the proposed protective role of choline in GDM pathology.

## 8. Conclusions

Several one-carbon metabolites have been implicated in risk for GDM. Specifically, increasing concentrations of circulating folate [7,11,14,15,16] and homocysteine [113,120,121,122,127], and decreasing vitamin B12 [14,15,111,112,115] and betaine [175,176,177] associate with increased GDM risk. Excess intake of FA has emerged as a key risk factor in GDM development. While the mechanism is largely unknown, some research suggests high FA may have a direct effect on pancreatic β-cell signalling [44,45]. There is growing evidence that dysregulation of several aspects of the one-carbon metabolic pathway may be at play (Summarised in Figure 3). Proposed interactions include a limited capacity to incorporate excess FA into the one-carbon metabolic pathway which can affect downstream transcription and methylation events, perturbations to the function and activity of one-carbon metabolism enzymes, notably MTR, TS and MTHFR. Alternatively, the relationship between excess FA and GDM risk may not be mediated by one-carbon metabolism and may instead result from direct adverse effects of circulating uFA which appears in circulation in response to saturated DHFR capacity. However, this is not well-established in the literature. Currently, the role of excess FA in dysregulation of aspects of one-carbon metabolism relies largely on limited in vitro research warranting further studies. Nevertheless, one-carbon metabolism is closely entwined with GDM risk. Given rising rates of GDM around the world and very widespread global FA food fortification, in the absence of vitamin B12 fortification, further research is urgently needed to elucidate the mechanisms by which perturbations of one-carbon metabolism, including high circulating uFA, contribute to GDM pathogenesis.

## Figures and Tables

**Figure 1 nutrients-14-03930-f001:**
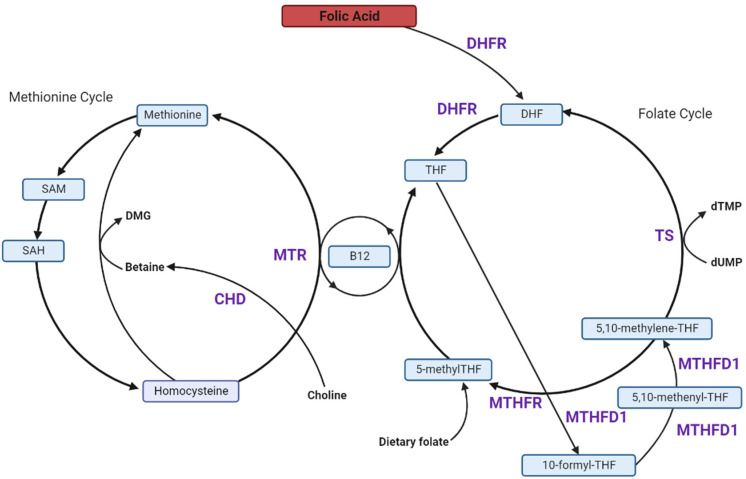
Overview of one-carbon metabolism. Folic acid (FA) is reduced via dihydrofolate reductase (DHFR) to dihydrofolate (DHF) and sequentially tetrahydrofolate (THF) [58]. THF is interconverted to intermediate metabolites 10-formyltetrahydrofolate (10-formylTHF), 5,10-methenyltetrahydrofolate (5,10-methenylTHF and 5,10-methylenetetrahydrofolate (5,10-methyleneTHF), Methylenetetrahydrofolate dehydrogenase (MTHFD1) regulates conversion of THF. After conversion of THF to 5,10-methyleneTHF), a substrate of methylenetetrahydrofolate reductase (MTHFR), 5,10-methyleneTHF can be used in the conversion of deoxyuridine monophosphate (dUMP) to deoxythymidine monophosphate (dTMP) via thymidylate synthase (TS). Alternatively, MTFHR converts 5,10-methyleneTHF to 5-methylTHF. 5-methylTHF is used for homocysteine re-methylation to methionine and is reliant on vitamin B12 (B12)-dependent methionine synthase (MTR). Methionine is converted to S-adenosylmethionine (SAM), a methyl donor in methylation reactions, and sequentially to S-adenosylhomocysteine (SAH), a substrate of homocysteine re-methylation. Alternatively, betaine derived from choline catalysed by choline dehydrogenase (CHD), can be used as a methyl donor in homocysteine re-methylation in a folate-independent manner. After donating a methyl group, betaine becomes dimethylglycine (DMG) Adapted from [16].

**Figure 2 nutrients-14-03930-f002:**
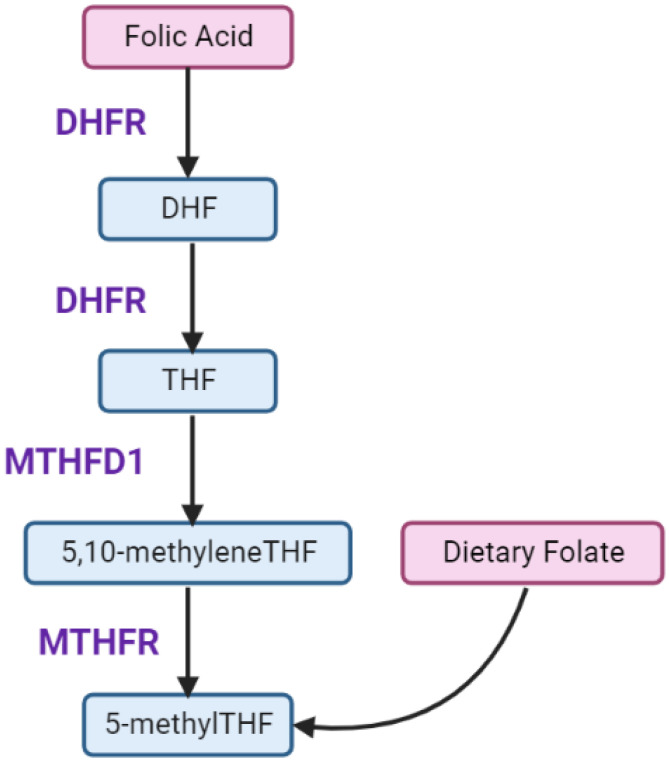
Folic Acid Metabolism. Folic acid (FA) requires reduction by dihydrofolate reductase (DHFR) to dihydrofolate (DHF) and sequentially tetrahydrofolate (THF). THF is then converted to 5,10-methyleneTHF and 5-methylTHF, dependent on methylenetetrahydrofolate dehydrogenase 1 (MTHDF1) and methylenetetrahydrofolate reductase (MTHFR) function.

**Figure 3 nutrients-14-03930-f003:**
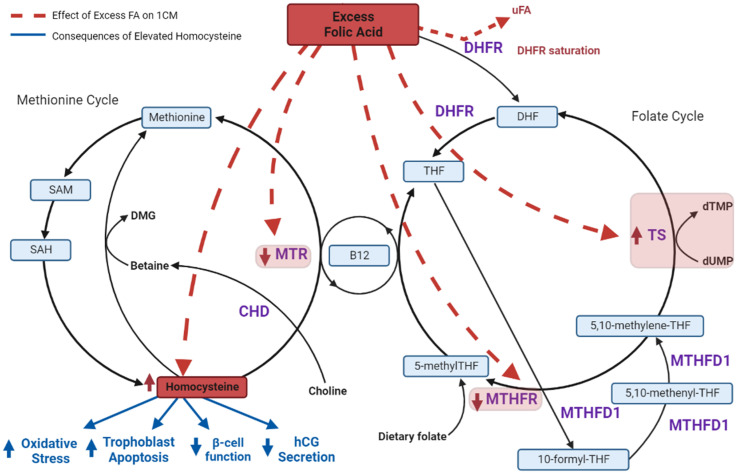
Overview of the effects of supraphysiological FA on one-carbon metabolism. Excess FA has demonstrated reduced methylenetetrahydrofolate reductase (MTHFR) activity in humans and mice favouring the thymidylate synthase (TS) cycle at the expense of methionine synthase (MTR) activity in *C. elegans* and elevated homocysteine in *C. elegans*, human choriocarcinoma BeWo and JEG3 cells.

**Table 1 nutrients-14-03930-t001:** Summary of research on the relationship between folate and GDM.

Authors	Study Country	Pregnancy Status (*n)*	Weeks’ Gestation	Measure	Results
Zhu et al., 2016 [7]	China	Non-GDM (1689) vs. GDM (249)	<12	FA supplementation	aOR: 2.25 95% CI: 1.35–3.76
Cheng et al., 2019 [9]	China	Non-GDM (853) vs. GDM (97)	≥3 months pre-conception	FA supplementation	aRR: 1.72 95% CI: 1.17–2.53, *p* < 0.01
Huang et al., 2019 [10]	China	Non-GDM (293) vs. GDM (33)	16–18	FA supplementation	aOR: 3.45 95% CI: 1.01–11.8, *p* < 0.05
Chen et al., 2021 [8]	China	Non-GDM (878) vs. GDM (180)	9–13	FA supplementation	aOR: 1.73 95% CI: 1.19–2.53, *p* = 0.004
				RBC Folate	aOR: 1.58 95% CI: 1.03–2.41, *p* = 0.033
Xie et al., 2019 [11]	China	Uncomplicated (1890) vs. GDM (392)	19–24	RBC folate	RR per 1-SD increase: 1.16 95% CI 1.03–1.30, *p* = 0.012
Liu et al., 2020 [12]	China	Non-GDM (299) vs. GDM (67	<12	RBC Folate	aOR: 2.473 95% CI: 1.013–6.037, *p* = 0.047
Li et al., 2019 [14]	China	Uncomplicated (316) vs. GDM (90)	24–28	Serum folate	OR: 1.98 95% CI: 1.00–3.90, *p* = 0.049
Saravanan et al., 2021 [15]	UK	Uncomplicated (3702) vs. GDM (526)	12.5 ± 1.4	Serum folate	aRR: 1.11 95% CI: 1.036–1.182, *p* = 0.002
Jankovic-Karasoulos et al., 2021 [16]	Australia and New Zealand	Uncomplicated (111) vs. GDM (33)	15 ± 1	Serum folate	mean ± SD (nmol/L): 31.9 ± 11.2 vs. 37.6 ± 8.0, *p* = 0.007) aOR: 1.22 (0.93–1.59), *p* = 0.149

**Table 2 nutrients-14-03930-t002:** Summary of research on the relationship between circulating vitamin B12 and GDM.

Authors	Study Country	Pregnancy Status (*n*)	Weeks’ Gestation	Measure	Results
Sukumar et al., 2016 [115]	UK	B12-deficient < 150 pmol/L (90) vs. B12-replete > 150 pmol/L (254)	26.9 ± 5.3	WHO 1999 GDM criteria	OR: 2.59 95% CI 1.35–4.98, *p* = 0.004. aOR: 2.05, 95% CI: 1.03–4.10, *p* = 0.04
Saravan et al., 2021 [15]	UK	B12-deficient < 220 pmol/L (1790) vs. B12-replete > 220 pmol/L (2530)	12.5 ± 1.4	IADPSG-GDM	aRR: 1.383, 95% CI 1.157–1.652, *p* = 0.0004
		Uncomplicated (3687) vs. GDM (633)	12.5 ± 1.4	Serum B12	aRR: 0.856, 95% CI: 0.786–0.933, *p* = 0.0004
				B12 tertile 1 + folate tertile 3	aRR: 1.742 95% CI: 1.226–2.437, *p* = 0.003
Li et al., 2019 [14]	China	Uncomplicated (110) vs. GDM (27)	24–28	Serum folate:B12 ratio 26.67–41.03	aOR: 1.53 95% CI: 0.79–2.97, *p* = 0.211
		Uncomplicated (93) vs. GDM (43)	24–28	Serum folate:B12 ratio ≥ 41.03	aOR: 3.08 95% CI: 1.63–5.83, *p* = 0.001
Lai et al., 2018 [112]	Singapore	Folate Tertile 1 (Ref) (193) vs. Folate Tertile 2 (164)	26–28	WHO 1999 GDM criteria	aOR: 1.94 95% CI: 1.04–3.62, *p* = 0.036
		Folate Tertile 1 (Ref) (193) vs. Folate Tertile 3 (156)	26–28	WHO 1999 GDM criteria	aOR: 1.97 95% CI: 1.05–3.68, *p* = 0.034
Krishnaveni et al., 2009 [111]	India	Folate ≤ 21.3 nmol/L (129) vs. Folate > 21.3–45.4 nmol/L (114) and Folate > 45.4 nmol/L (91)	30	GDM Carpenter–Coustan criteria [116]	5.4%, 10.5%, 10.9% (Tertile 1, 2, and 3, respectively), *p* = 0.04

WHO, World Health Organisation; IADPSG, International Association of the Diabetes and Pregnancy Study Groups; GDM, gestational diabetes mellitus; aOR, adjusted odds ratio; CI, confidence interval aRR, adjusted risk ratio.

**Table 3 nutrients-14-03930-t003:** Summary of research on the relationship between circulating homocysteine and GDM.

Authors	Study Country	Pregnancy Status (*n)*	Weeks’ Gestation	Hcy Measure	Results
Tarim et al., 2004 [120]	Turkey	Normoglycemic ≤ 7.5 nmol/L, 1 h-50 g glucose (210) vs. glucose intolerant, >7.5 nmol/L glucose challenge, normal oGTT (66) vs. GDM (28)	24–28	Plasma 8 h fasting	Mean ± SD (μmol/L) Group 1: 4.80 ± 0.98 Group 2: 5.51 ± 1.08 Group 3: 5.70 ± 0.90 (*p* < 0.001)
Guven et al., 2006 [121]	Turkey	Normoglycemic ≤ 7.8 nmol/L, 1 h-50 g glucose (147) vs. glucose intolerant > 7.8 nmol/L glucose challenge, normal oGTT (46) vs. GDM (30)	24–28	Serum	Mean ± SD (μmol/L) Group 1: 7.4 ± 1.6 Group 2: 8.1 ± 2.5 Group 3: 9.0 ± 3.1, *p* < 0.01
Seghieri et al., 2003 [122]	Italy	Non-GDM (78) vs. GDM (15)	24–28	Serum	Mean ± SD (μmol/L) Control: 4.45 ± 1.52 GDM: 5.88 ± 2.26, *p* = 0.003
Tarim et al., 2006 [123]	Turkey	Non-GDM (40) vs. GDM (30)	24–28	Plasma	Mean ± SD (μmol/L) Control: 5.03 ± 0.91 GDM: 5.96 ± 1.70 *p* = 0.027
Davari-Tanha et al., 2008 [124]	Iran	Non-GDM (40) vs. GDM (40)	24–28	Plasma 8 h fasting	Mean ± SD (μmol/L) Control: 5.05 ± 1.1 GDM: 7.8 ± 1.6 *p* < 0.0001
Atay et al., 2014 [125]	NS	Uncomplicated (38) vs. GDM (37)	24–28	Serum 12 h fasting	Mean ± SD mmol/l) Control: 5.91 ± 3.87 GDM: 9.57 ± 4.46 *p* < 0.001
Deng et al., 2020 [126]	China	Non-GDM (350) vs. GDM (346)	24–28	Plasma	Mean ± SD (μmol/L) Control: 6.17 ± 1.29 GDM: 6.61 ± 1.32 *p* = 0.001
Idzior-Waluś et al., 2008 [127]	Poland	Non-GDM (17) vs. GDM (44)	26–32	Serum	Mean ± SD (μmol/L) Control: 7.4 ± 1.1 GDM 8 ± 2.0 NS
Radzicka et al., 2019 [113]	Poland	Uncomplicated (19) vs. GDM (60)	24–28	Serum	Mean ± IQR (μmol/L) Control: 8.02 ± 2.27 GDM: 7.41 ± 2.61 (NS)
López-Quesada et al., 2005 [129]	Spain	Normoglycemic ≤ 7.8 nmol/L, 1 h-50 g glucose (190) vs. Glucose intolerant (18) > 7.8 nmol/L glucose challenge, normal oGTT vs. GDM (17)	34	Plasma fasting	Median ± SD (μmol/L) Group 1: 6.6 ± 2.0 Group 2: 5.0 ± 1.7 Group 3 6.8 ± 2.7
Akturk et al., 2010 [131]	Turkey	Normoglycemic (69) vs. GDM (54)	32–39	Plasma	Mean ± SEM (μmol/L) Control: 5.62 ± 0.34 GDM: 5.20 ± 0.30
Mascarenhas et al. 2014 [132]	India	Normoglycemic (83) vs. GDM (7)	8–12	Serum overnight fasting	Mean (μmol/L) Control: 14.41 ± 7.98 GDM: 15.66 ± 7.61 *p* = 0.6312

oGTT, oral glucose tolerance test; GDM, gestational diabetes mellitus; NS, not stated; SD, standard deviation; IQR, interquartile range; SEM, standard error of mean.

**Table 4 nutrients-14-03930-t004:** Summary of research on the relationship between betaine and GDM.

Authors	Study Country	Pregnancy Status (*n)*	Weeks’ Gestation	Measure	Results
Huo et al., 2019 [175]	China	Uncomplicated (243) vs. GDM (243)	Median: 10 (IQR: 9–11)	Serum betaine	Mean (IQR) (nmol/mL) Control: 290.4 (244.2–378.8) GDM: 229.7 (195.6–279.9), *p* < 0.0001
		Betaine ≤ 200 nmol/mL (90) vs. Betaine > 200 nmol/mL (396)	Median: 10 (IQR: 9–11)	GDM (WHO 2013 criteria)	OR: 5.00 95% CI: 2.76–9.07, *p* < 0.0001 aOR: 4.88 95% CI 2.51–9.50, *p* < 0.0001
Gong et al., 2021 [176]	China	Betaine Tertile 1 (62) vs. Betaine Tertile 2 (63) vs. Betaine Tertile 3 (62)	5.4–11.4	IADPSG-GDM	aRR: 0.41 (95% CI: 0.19– 0.86, *p*-trend = 0.015
Barzilay et al., 2018 [177]	Canada	Uncomplicated (296) vs. GDM (18)	12–16	Plasma betaine	Mean ± SD (μmol/L): 13.4 ± 4.1 vs. 12.1 ± 2.4, *p* = 0.15
		Uncomplicated (278) vs. GDM (16)	37–42	Plasma betaine	Mean ± SD (μmol/L): 10.4 ± 2.8 vs. 10.3 ± 2.2, *p* = 0.92
		Uncomplicated (252) vs. GDM (14)	28–42	Cord blood plasma betaine	Mean ± SD (μmol/L): 21.2 ± 4.7 vs. 18.5 ± 3.9, *p* = 0.02

GDM, gestational diabetes mellitus; IQR, interquartile range; WHO, World Health Organisation; aOR, adjusted odds ratio; CI, confidence interval; IADPSG, International Association of the Diabetes and Pregnancy Study Groups; aRR, adjusted risk ratio; SD, standard deviation; RBC, red blood cell.

## Data Availability

Not applicable.

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
