# Peer review of "High Folate, Perturbed One-Carbon Metabolism and Gestational Diabetes Mellitus"

_nutrients, 2022, doi:10.3390/nu14193930_

Round 1
Reviewer 1 Report
The Authors deal with a very interesting topic The literature has been carefully studied and well discussed
Reviewer 2 Report
Williamson et al. provided a detailed and comprehensive review on the one carbon nutrients in GDM. The authors provided an extensive list for some potential mechanisms from studies that investigated one carbon disturbances, indicating the influence in health outcomes. The authors also describe the methyl folate trap in GDM with great detail (PMID: 34210607).
The article lacks an indication of metabolic subcellular localization. One carbon metabolism, especially the production of betaine from choline, rely on effective mitochondria (PMID: 23906661). GDM compromises ova mitochondria (PMID: 32945868). Including these details will include important context to otherwise trivial disturbances in one carbon nutrients.
The article also lacks a clear indication of one-carbon metabolites and their direct influence on the disturbed mechanism of action of insulin in the insulinopathy. Detailing or proposing a mechanism of direct action of one carbon nutrients to the impaired insulin pathway could be included to improve clarity and relevance. (PMIDs: 33498674, 34754676, and 33091596)
Overall, the paper is informative and includes a good review of the current literature. Possibly try to re-render the figures as they seemed a bit pixelated.
